# Pharmacokinetics and Endocrine Effects of an Oral Dose of D-Pinitol in Human Fasting Healthy Volunteers

**DOI:** 10.3390/nu14194094

**Published:** 2022-10-01

**Authors:** Juan A. Navarro, Caridad Díaz, Juan Decara, Dina Medina-Vera, Antonio J. Lopez-Gambero, Juan Suarez, Francisco Javier Pavón, Antonia Serrano, Antonio Vargas, Ana Luisa Gavito, Oscar Porras-Perales, Jesús Aranda, Francisca Vicente, Carlos Sanjuan, Elena Baixeras, Fernando Rodríguez de Fonseca

**Affiliations:** 1Laboratorio de Medicina Regenerativa, Unidad de Gestión Clínica de Salud Mental, Instituto de Investigación Biomédica de Málaga (IBIMA), Hospital Regional Universitario de Málaga, 29010 Málaga, Spain; 2Facultad de Medicina, Campus de Teatinos s/n, Universidad de Málaga, 29010 Málaga, Spain; 3Department of Screening & Target Validation, Fundación MEDINA, 18016 Granada, Spain; 4Unidad de Gestión Clínica del Corazón—CIBERCV (Enfermedades Cardiovasculares), Hospital Universitario Virgen de la Victoria, 29010 Málaga, Spain; 5Facultad de Ciencias, Campus de Teatinos s/n, Universidad de Málaga, 29010 Málaga, Spain; 6Departamento de Anatomía Humana, Medicina Legal e Historia de la Ciencia, Facultad de Medicina, Universidad de Málaga, 29010 Málaga, Spain; 7Euronutra S.L. Calle Johannes Kepler, 3, 29590 Málaga, Spain; 8Departamento de Bioquímica y Biología Molecular, Facultad de Medicina, Universidad de Málaga, 29010 Málaga, Spain

**Keywords:** D-Pinitol, inositol, diabetes, carob fruit, ghrelin, insulin, insulin resistance, pituitary hormones

## Abstract

The present study characterizes the oral pharmacokinetics of D-Pinitol, a natural insulin mimetic inositol, in human healthy volunteers (14 males and 11 females). D-Pinitol absorption was studied in (a) subjects receiving a single oral dose of 15 mg/kg (*n* = 10), or (b) 5 mg/kg pure D-Pinitol (*n* = 6), and (c) subjects receiving D-Pinitol as part of carbohydrate-containing carob pods-derived syrup with a 3.2% D-Pinitol (Dose of 1600 mg/subject, *n* = 9). The volunteers received a randomly assigned single dose of either D-Pinitol or carob pod-derived syrup. Blood samples were collected at 0, 15, 30, 45, 60, 90, 120, 180, 240, 360 and 1440 min after intake. Plasma concentration of D-Pinitol was measured and pharmacokinetic parameters obtained. The data indicate that when given alone, the oral absorption of D-Pinitol is dose-dependent and of extended duration, with a Tmax reached after almost 4 h, and a half-life greater than 5 h. When the source of D-Pinitol was a carob pods-derived syrup, Cmax was reduced to 40% of the expected based on the data of D-Pinitol alone, suggesting a reduced absorption probably because of competition with monosaccharide transport. In this group, Tmax was reached before that of D-Pinitol alone, but the estimated half-life remained the same. In the D-Pinitol groups, plasma concentrations of glucose, insulin, glucagon, ghrelin, free fatty acids, and pituitary hormones were additionally measured. A dose of 15 mg/kg of D-Pinitol did not affect glucose levels in healthy volunteers, but reduced insulin and increased glucagon and ghrelin concentrations. D-Pinitol did not increase other hormones known to enhance plasma glucose, such as cortisol or GH, which were surprisingly reduced after the ingestion of this inositol. Other pituitary hormones (gonadotropins, prolactin, and thyroid-stimulating hormone) were not affected after D-Pinitol ingestion. In a conclusion, D-Pinitol is absorbed through the oral route, having an extended half-life and displaying the pharmacological profile of an endocrine pancreas protector, a pharmacological activity of potential interest for the treatment or prevention of insulin resistance-associated conditions.

## 1. Introduction

Dietary inositols have been proposed as a natural solution to improve the metabolic disturbances associated with insulin resistance [1,2]. Their beneficial effects have been attributed to their pharmacological activity as insulin sensitizers, improving insulin receptor-assciated signaling cascade. Traditionally the pharmacological action of dietary inositols was related to their incorporation into inositol phosphoglycans (IPG), complex molecules composed of one inositol, an aminosugar, and several phosphate residues. IPG species act as relevant post-receptor mediators of insulin action [3,4]. These IPG-mediated actions require repeated administration of the inositols in the diet to allow the organisms to incorporate those cyclitols into the IPG structure. The importance of an external supplementation of these inositols is relevant in diabetes, where the renal clearance of inositols is enhanced because of the competition of excessive urinary glucose with inositol uptake by renal tubules [5,6]. Among the several inositols studied for their potential use against insulin resistance we can cite myoinositol, D-Chiro-inositol (DCI), and D-Pinitol (DPIN, 3-0-methyl-d-chiro-inositol), being the latter considered a potential dietary source of DCI since it can be demethylated in the acid media of the stomach. D-Pinitol is found in high concentrations in several medicinal plants (i.e., Bougainvillea spectabilis), carob pods (Ceratonia siliqua), or legumes such as soybean (Glycine max) [7]. D-Pinitol can also be incorporated into inositolphosphoglycans (PGI), as it is the insulin’s second messenger INS-2, thus having similar effects on insulin signaling as IPG containing D-Pinitol [8,9].

In addition to the effects derived from the dietary administration of inositols, recent studies have identified quick metabolic and hormonal responses associated with acute oral administration of these inositols, suggesting the existence of mechanisms not derived from the incorporation of inositols into IPG. From these studies, DPIN emerges as a molecule capable of improving glucose handling in oral carbohydrate load tests, both in humans and preclinical models [10]. Detailed studies in rodents, demonstrated that the oral administration of DPIN resulted in plasma elevation of this inositol, associated with maintenance of glycemia stability, despite a clear decrease of insulin secretion [11]. This dual effect, which might help to protect the endocrine pancreas from excessive insulin secretion, was explained based on a) enhanced ghrelin secretion that inhibits the pancreatic release of insulin and b) enhanced glucagon release that sustains glycaemia through regulation of glycolysis/neoglucogenesis processes in the liver [11]. In addition, DPIN directly activated insulin signaling in the hypothalamus, despite the drop in plasma insulin concentrations, suggesting a direct action on post-receptor insulin signaling processes [12]. Moreover, in vitro administration of DPIN can enhance glucose uptake in the myoblast cell line L6 [8], acting as glucose, an effect that might also explain why hyperglycaemia is not seen despite the opposite changes in insulin and glucagon concentrations in plasma observed in the present experiment. 

Overall, these effects made DPIN a molecule to be considered an active ingredient for healthy food preparations. However, except for a time frame-limited assay with either D-Pinitol [13] or a carob fruit-derived syrup [10], there is no information on the pharmacokinetics of DPIN in humans, and on its impact on these ghrelin-insulin-glucagon actions in healthy subjects that might confirm the activity described in rodents in preclinical assays. The present investigation addressed those questions in human healthy volunteers, as well as the potential interference that the presence of dietary glucose or fructose might exert on D-Pinitol absorption.

## 2. Materials and Methods

### 2.1. Study Design

The main aim of the study was first to investigate the oral absorption of D-Pinitol when given alone, obtaining the pharmacokinetic parameters Cmax, Tmax, and half-life after monitoring plasma concentrations of this inositol at several points after the ingestion. A dose of 15 mg/kg (equivalent to 1050 mg in a subject of 70 kg) was selected, on the basis of previous experimental findings in rodents. A pilot experiment using a low dose of 5 mg/kg was also carried out to obtain absorption data with low doses for monitoring the efficacy of absorption. A secondary aim was to analyze whether the presence of carbohydrates (glucose/fructose) might interfere with the absorption of D-Pinitol. To this end, human healthy volunteers received an oral administration of a carob pod syrup containing 50 g of total carbohydrates dose (InnoSweet ^®^) containing 45.6% glucose, 47.3% fructose, 0.5% sucrose, and 3.2% D-Pinitol (50 g of syrup contains a dose of D-Pinitol of 1600 mg or 22.8 mg/kg body weight in a person weighing 70 kg). A third aim was to analyze whether the acute administration of D-Pinitol was capable of modifying glycemia and free fatty acid plasma concentrations, as well as altering glucose-modifying hormones such as insulin, glucagon, cortisol, growth hormone, and ghrelin, as it was previously described in rodents [11]. Since D-Pinitol can be converted to D-Chiroinositol that is capable of modifying the function of the hypothalamus-hypophysis-gonadal axis, we also monitored gonadotrophins (LH and FSH) and additional pituitary hormones (Prolactin (PRL), and Thyroid-stimulating hormone (TSH)).

To achieve the aims described above, we designed a randomized trial using D-Pinitol (3-0-methyl-d-chiro-inositol, 98% purity), or a carob pods-derived syrup (InnoSweet ^®^) that was generously provided by Euronutra (https://www.euronutra.com/ (accessed on 8 July 2022), Málaga, Spain). Subjects were randomly assigned to one of the following treatments: (A) 10 subjects received pure D-Pinitol (15 mg/kg body weight) prepared in 100 mL of drinking water; (B) 6 subjects received pure D-Pinitol (5 mg/kg body weight) prepared in 100 mL of drinking water, and finally (C) 9 subjects received 50 g of carbohydrates from a natural carob pod-derived syrup (InnoSweet ^®^) containing 45.6% glucose, 47.3% fructose, 0.5% sucrose and 3.2% D-Pinitol (equivalent to a dose of D-Pinitol of 1600 mg or 22.8 mg/kg body weight in a person weighing 70 kg), and diluted with 100 mL of drinking water. Sub-studies A and C were designed to analyze the pharmacokinetics of D-Pinitol in the absence (A) or presence (C) of additional carbohydrates in the solution administered, that might interfere with the absorption. Sub-studies A and B were designed to analyze the dose effect on absorption as well as the main effects on glycemia and plasma concentrations of insulin and glucagon hormones. 

Concerning the sample size, we calculated it using the Gpower program, version 3.1.9.2., taking into consideration that the expected variations in plasma D-Pinitol concentrations might exceed 3-fold in between time points. Recruitment of male and female volunteers was carried out to reach the calculated sample size, as described in Table 1 and the statistics section.

### 2.2. Subjects

In this case, 25 healthy volunteers were recruited among healthy clinical and laboratory staff of the Laboratory of Neuropsychopharmacology of the Regional University Hospital of Málaga. The inclusion criteria for all subjects were (a) age range of 18–65 years, (b) the presence of a baseline capillary blood glucose < 5.6 nmol/L (100 mg/mL) measured with a glucose oxidase method after overnight fasting, (c) the absence of obesity (BMI > 30) and the absence of a diagnosed/treated metabolic disease. Subjects were interviewed for present or past diagnosis of diabetes (Glucose > 7.8 mmol/L after a standard glucose load), hypertryglyceridemia, or hypercholesterolemia under treatment, as well as for a clinical record of past endocrine disorders including thyroid gland dysfunction or treatment with glucocorticoids. Based on the inclusion criteria, exclusion criteria were pregnancy or lactation, fasting glycemia > 5.6 mmol/L (100 mg/dL), and fasting insulinemia determined by ELISA > 25 mIU/L. diabetes or medication known to interfere with carbohydrate metabolism. Any of the women included used contraceptive medication. The main data of men and women participants can be found in Table 1. The study was conducted under the guidelines of the Declaration of Helsinki, and the research protocol was approved by the Ethics Committee of the Regional University Hospital of Málaga, under protocol number P18-TP-5194. 

### 2.3. Blood Sample Collection, Plasma Preparation, and Glucose Concentration Monitoring

After 12 h overnight fasting, a catheter was inserted into the antecubital vein of each subject, blood was extracted at baseline (minute 0; while still fasting), and immediately after (within 5 min) oral intake of the assigned doses of either DPIN (15 mg/kg) or the natural syrup. Next, blood samples were collected at 0, 15, 30, 45, 60, 90, 120, 180, 240, 360 and 1440 min after intake. Individuals were allowed to eat 6 h after the first blood sample. All-time points were used for measuring DPIN concentrations. Time points between 0 min (basal) and 180 min post DPIN(15 mg/kg) intake were used for measuring glucose, free fatty acids, glucagon, and insulin. Time points 0, 120 and 180 min post-D-Pinitol intake were used for the analysis of prolactin, thyroid-stimulating hormones, and gonadotropins. Time points of 0, 60, and 120 min post DPIN intake were used for measuring both, growth hormone and cortisol concentrations. In the case of the DPIN dose of 5 mg/kg dose, blood was withdrawn only at 0, 60, and 120 min post-DPIN ingestion to compare with the absorption observed with higher doses. All the blood samples were collected in Vacutest tubes (Vacutest Kima S.r.l., Arzergrande, PD, Italy, cat. number: #13560), centrifuged at 2000× *g* for 10 min at 4 °C, and the plasma was kept at −80 °C for further biochemical analysis.

Right after the blood was extracted from the subjects, plasma was obtained by centrifugation and plasma glucose concentrations were measured with a commercially available glucometer (AccuCheck, Roche, Germany). 

### 2.4. Quantitation of Pinitol: Liquid Chromatography-Mass Spectrometry Method

#### 2.4.1. Mass Spectrometry Method

A quantitation method for D-Pinitol in human plasma using liquid chromatography coupled to mass spectrometry (LC-MS/MS) was validated. Analysis was carried out using an Agilent 1290 (Agilent Technologies, Santa Clara, CA, USA) liquid chromatographer and an Api4000 triple quadrupole mass spectrometer (SCIEX). Quantitation was obtained using the multiple-reaction monitoring (MRM) mode of the transitions at m/z 195.2→109.0 (quantitative) and 195.2/80.0 (qualitative) for D-Pinitol both with collision energy to 20 Ev, and *m*/*z* 261.1→205.1 for the internal standard (IS), see Figure 1A,B. IS was provided by MEDINA.

Mass spectrometry conditions consisted of decluster potential (DP): 10 eV, GS1 and GS2: 45 psi, Tª: 600 °C and Ions spray: 5500 Ev. Chromatographic conditions were the following, the mobile phase (MP) consisted of 0.1% formic acid−[water:AcN] [90:10] (MP A) and 0.1% formic acid−[AcN:water] [90:10] (MP B). The gradient program was applied as follows: t = 0−0.5 min 95% MP B; t = 4.50–6.70 min 25% MP B; t = 6.80 95% MP B, t = 6.80–9:00 min 95% MP B. The flow rate was 0.3 mL/min and the run time was 9 min. The injection volume was 5 μL. The chromatographic column used was X bridge BEH amide (waters) with dimension 2.1 × 100 mm and particle size 3.5 μm. The oven column was maintained at 30 °C. The method performance was validated according to the FDA recommendations proposed in 2018 for selectivity, sensitivity, matrix effect, linearity, precision, accuracy, and recovery of plasma samples [14].

#### 2.4.2. Sample Preparation

An aliquot of 50.0 μL plasma was taken and 2% TFA plus 150 μL of cold methanol containing the internal standard was added. After vortex mixing for one minute, the samples were centrifuged for 15 min at 13,300 rpm. The temperature of the centrifuge was set at 4 °C. An aliquot of 160.0 μL of the supernatant was transferred to a vial for evaporation. The samples were reconstituted in 100 μL water/acetonitrile (80/20) and 0.1% ammonia (20%) for LCMS analysis.

#### 2.4.3. Pharmacokinetic Study

All pharmacokinetic parameters were estimated from the mean concentration values (*n* = 5–10) determined at each time point and were evaluated by noncompartmental analysis using PKSolver (Version 2.0) [15].

### 2.5. Quantitation of Glucose and Plasma Hormone Concentrations

Glucose homeostasis-related hormones: plasma levels of hormones regulating glucose homeostasis were determined by the Enzyme-Linked ImmunoSorbent Assay (ELISA) method using commercial kits: insulin (EMD Millipore Corporation, Billerica, MA, USA, cat. number: #EZHI-14K, intra-assay coefficient of variation of 4.86; inter-assay coefficient of variation of 5.73), glucagon (Elabscience Biotechnology Inc., Wuhan, Hubei, China, cat. number: #E EL H2237, intra-assay coefficient of variation of 5.36; inter-assay coefficient of variation of 7.95) and active ghrelin (Kamiya Biomedical Company, Seattle, WA, USA, cat. number: #KT-364, intra-assay coefficient of variation of 6.62; inter-assay coefficient of variation of 25.31). 

Free fatty acids plasma concentration was measured using a commercial kit (Sigma-Aldrich, Saint Louis, MO, USA, cat. number: MAK044) according to the manufacturer’s instructions. All serum samples were assayed in duplicate within one assay, and results were expressed in terms of the particular standard metabolite. 

Insulin resistance was evaluated according to the Homeostatic Model Assessment for Insulin Resistance (HOMA-IR index), calculated using the following formula: HOMA-IR = ((fasting plasma insulin [μIU/mL] × fasting blood glucose [mmol/L])/22.5). Homeostatic assessment of β-cell function index (HOMA β), was calculated using the formula: HOMA β = [(360 × fasting serum insulin in μU/mL)/(fasting blood glucose in mg/dL − 63)] [16]. With the previously insulin and glucagon values obtained we calculated the glucagon/insulin ratio.

Plasma levels of pituitary hormones: follicle-stimulating hormone (FSH, expressed in mU/mL; intra-assay coefficient of variation of 7; inter-assay coefficient of variation of 8), human growth hormone-1 (GH, expressed in ng/mL; intra-assay coefficient of variation of 6; inter-assay coefficient of variation of 7), luteinizing hormone (LH, expressed in mU/mL; intra-assay coefficient of variation of 4; inter-assay coefficient of variation of 6), prolactin (PRL, expressed in ng/mL; intra-assay coefficient of variation of 8; inter-assay coefficient of variation of 10.4) and thyroid stimulating hormone (TSH, expressed in µU/mL; intra-assay coefficient of variation of 7; inter-assay coefficient of variation of 8.1) were determined by multiplex immunoassay system, using a commercial kit: Bio-Plex Pro™ RBM Human Hormone Panel 1 (Bio-Rad, Hercules, CA, USA, cat. number: #171AHR1CK). The plate was run on a Bio-Plex MAGPIX™ Multiplex Reader with Bio-Plex Manager™ MP Software (Luminex, Austin, TX, USA). Hormone detection limits were: 0.11 mU/mL (FSH), 0.0076 ng/mL (GH), 0.061 mU/mL (LH), 0.022 ng/mL (PRL) and 0.012 µU/mL (TSH).

Plasma cortisol levels were determined using a commercial kit (Cayman Chemical Company, Ann Arbor, MI, USA, cat. Number: 500360; intra-assay coefficient of variation of 17.6; inter-assay coefficient of variation of 18.7) according to the manufacturer’s instructions, plasma preparation was performed as recommended by the kit, and results were expressed in terms of the standards provided. Absorbances obtained after reading the plate were analyzed using a computer spreadsheet provided by the same producer, available at https://www.caymanchem.com/analysisTools/ELISA (accessed on 22 March 2022).

### 2.6. Statistical Analysis

The main variables of the study were the Cmax and the area under the curve (AUC) obtained when monitoring variations of plasma D-Pinitol levels at different times after oral intake. Since plasma concentrations at the different time points selected after oral intake of D-Pinitol can display more than a 3-fold variation, we selected a restrictive size of the effect of 2. That gives a sample size of 7 subjects per group for comparing D-Pinitol levels versus time after ingestion of either D-Pinitol or Carob Syrup. Since we wanted to have a representation of men and women, we increased the numbers to 10 and 9 respectively. For the dose of 5 mg/kg, we could only recruit 6 out of the 7 subjects, so we decided to present these data as Appendix A, but we did not perform a full analysis with this sub-study; thus, we recruited a total of 25 subjects.

Graph-Pad Prism 8.0 software (GraphPad Software, Inc., San Diego, CA, USA) was used to analyze the data. Values are represented as mean ± standard error of the mean (SEM) for each experimental group, according to the assay. The significance of differences within and between groups was evaluated by a one-way or two-way (depending on the assay) analysis of variance (ANOVA) followed by a posthoc test for multiple comparisons. A *p*-value ≤ 0.05 was considered to be statistically significant. (* = *p* < 0.05; ** = *p* < 0.01; *** = *p* < 0.001).

## 3. Results

### 3.1. Patient Recruitment and Basal Glucose Homeostasis Parameters

Table 1 shows the characteristics of the volunteers recruited for (a) the D-Pinitol sub-study (Including both D-Pinitol doses of 15 and 5 mg/kg of body weight), and (b) the carob pod syrup substudy. There were no differences between both groups in any of the variables (age, weight, body mass index (BMI), plasma glucose, plasma insulin, or HOMA-IR index of insulin resistance). These parameters were found to be in the normal range, so all the recruited subjects were used for the subsequent analysis. The analysis of differences between both sexes revealed, as expected, that females displayed lower body weight (F(1,21) = 11.3, *p* < 0.003, 2-way ANOVA) which led to a reduced BMI in the females. 

### 3.2. Method Validation for D-Pinitol Quantitation

Calibration curves were linear over the concentration range of 0.31–20 μg/mL. The correlation coefficient (r2) was 0.9963 for the calibration curve in the analysis (Figure 1C). Precision, accuracy, matrix effect recovery, and stability were according to criteria and conformed to the guidelines for bioanalytical method validation [14].

We have developed an analytical method for D-Pinitol quantitation using mass spectrometry with a lower limit of quantitation of 313 ng/mL and a lower limit of detection of 200 ng/mL. The measurement of cyclic polyol using mass spectrometry and HPLC-DAD is complicated due to poor ionization and response [17] respectively. However, our method showed good sensitivity and selectivity to quantifying D-Pinitol in plasma samples.

### 3.3. Pharmacokinetics Analysis of D-Pinitol Alone or as Part of a Natural Carob Syrup

Pharmacokinetics analysis of oral administration of DPIN was obtained by monitoring plasma concentration of this inositol at 0, 15, 30, 45, 60, 90, 120, 180, 240, 360 and 1440 min after the oral administration of (a) a 15 mg/kg body weight dose or (b) 50 g of carbohydrates from a natural syrup containing monosaccharides (45.6% glucose, 47.3% fructose), disaccharides (0.5% saccharose) and 3.2% D-Pinitol, (% weight) at a dose equivalent to 1.6 gr/subject of DPIN. The subjects were healthy middle-aged male and female volunteers that were fasting for 12 h. As shown in Figure 2A,B, fasting plasma DPIN concentrations were below the level of detection (minute 0). After supplementation, plasma DPIN became detectable as soon as 30 min after the ingestion and peaked (Tmax) at 240 min post-intake when DPIN was given alone and peaked (Tmax) at 90 min post-intake when it was given as a component of the natural carob-pod derived syrup. Absolute maximal concentrations (Figure 3A,B) of DPIN (Cmax), as well as half-life, were similar in both cases, despite the different doses administered (15 mg/kg body weight in males and females for DPIN alone, and 1600 mg in the case of the syrup, that corresponds to an estimated dose of 19 mg/kg body weight in males and 23.4 mg/kg body weight in the case of females). 

However, if we calculated a dose-normalized ratio for Cmax, using the following formula normalized ratio = Cmax (ng/mL)/Dose (mg/mL), we observed a reduction of more of 40% of the expected Cmax in the carob syrup group, suggesting that the presence of carbohydrates resulted on interference in D-Pinitol Absorption (Normalized Ratio for D-Pinitol = 1.73 ± 0.65; Normalized Ratio for Carob Syrup = 0.97 ± 0.56, Means ± Standard Deviation. Both groups were different (T-Student t = 2.72, df = 17, *p* < 0.01). Moreover, at each point analyzed, estimated plasma concentrations of DPIN in the first group showed less variability than in the case of the syrup. These three variables, different Tmax, more absorption variability, and the different relative Cmax values indicate that DPIN is more reliable incorporated into the bloodstream when it is administered alone, and that the presence of carbohydrates might interfere with the absorption. D-Pinitol concentrations were dosed-dependent since the administration of 5 mg/kg resulted in proportional reductions in the plasma levels monitored in this low dose of DPIN (Appendix A). Analysis of half-life indicates that clearance of DPIN in humans is much slower than in rats [11], in accordance with the lower rate of metabolism and renal clearance in humans when compared with rodents.

Finally, there were no correlations of D-Pinitol Cmax with BMI (r2 = 0.44, *p* (two-tailed) = 0.05). Mean peak plasma D-Pinitol values were similar in females (1527 + 589 ng/mL) than in male subjects (1562 + 178 ng/mL).

### 3.4. Effects of D-Pinitol on Plasma Glucose and Its Controlling Mechanisms

Neither the administration of 15 mg/kg, nor that oft 5 mg/kg (Appendix A), of DPIN, resulted in alterations of plasma glucose (Figure 4A. However, 15 mg/kg D-Pinitol resulted in a decrease in insulin secretion (F(7,72) = 2.9, *p* < 0.01, Figure 4B), slightly reducing the HOMA index of insulin resistance (F(7,64) = 2.6, *p* < 0.05, Figure 4C). HOMA β index was not affected (Figure 4D). D-Pinitol administration resulted in a time-dependent increase in glucagon (F(7,67) = 3.85, *p* < 0.005, Figure 4E). Finally, DPIN administration resulted in enhanced acylated ghrelin secretion (F(4,38) = 13.88, *p* < 0.001, Figure 4F). Acylated Ghrelin was found to be enhanced 60 after DPIN ingestion.

### 3.5. Effect of D-Pinitol on Pituitary Hormones and Cortisol

Growth hormone, and the pituitary hormone with a major impact on glucose levelswas the only pituitary hormone whose plasma concentration was affected (reduced) by the administration of DPIN (F(2,23) = 3.52, *p* < 0.05, Figure 5A). Neither Prolactin, TSH, LH nor FSH (Figure 6A–D) were affected by the intake of DPIN,. In addition, to monitor the impact of DPIN on glucocorticoids, a steroid hormone that rises glucose, we measured plasma cortisol. The concentration of this hormone was reduced by DPIN (F(2,21) = 7.38, *p* < 0.005, Figure 5B).

## 4. Discussion

The present study shows the pharmacokinetic profile of D-Pinitol when given orally as a pure food ingredient alone. Data reveals that after ingestion of DPIN, it follows an extended period of absorption that lasts almost four hours, decreasing thereafter. The present experimental design includes sampling times (6 and 24 h post-DPIN ingestion) that were not evaluated in previous studies monitoring plasma DPIN concentrations after oral intake. Those studies limited the analysis of DPIN concentrations to the first 4 h post-ingestion, complicating any estimation of pharmacokinetics properties of DPIN [10,13]. The present design allowed the calculation of DPIN half-life that reached five and a half hours. This long half-life value allows coverage of large postprandial and nocturnal periods, where the DPIN might exert its effects after a single administration. The maximal concentrations of DPIN (Cmax of 1392 ng/mL (7.17 μM)) observed in our experimental design, where the dose was adjusted to the weight of each subject, are in agreement with previous published using either DPIN alone (1 g of DPIN, yielding Cmax around 10 μM [13]) and those observed after ingestion of higher doses of DPIN in a natural syrup (Cmax of 1670 ng/mL (8.6 μM) for a dose of 2.5 g of DPIN [10]). It is important to highlight that both studies were carried out in the context of a glucose load experiment, where the presence of this monosaccharide might interfere with the absorptive processes of DPIN. This is what can be deduced from the Cmax of the study that used a 2.5 g of DPIN from a natural syrup (InnoSweet ^®^), whose value did not duplicate the Cmax of our study, where doses were in the range of 0.75–1.3 gr/person, depending on the weight of the subject. The small difference in the Cmax between both studies indicates that in the presence of carbohydrates, mainly monosaccharides, DPIN absorption is reduced, probably because of the competition of glucose and fructose for the inositol transporter [18]. We confirmed this finding in a parallel study using a natural syrup containing monosaccharides (45.6% glucose, 47.3% fructose, 0.5% saccharose, and 3.2% D-Pinitol, % weight) at a dose equivalent to 1.6 gr/subject of DPIN. While half-life was identical, Cmax was not different, reaching 1492 ng/mL, despite being the D-Pinitol dose used in the syrup (22.8 mg/kg in a person of 70 kg) 50% greater than de 15 mg/kg dose used in the study where D-Pinitol was given alone A pilot study (see Appendix A) with a dose of 5 mg/kg of DPIN indicated that the absorption was proportional to the dose administered since we did not observe an enhancement or lowering of the expected plasma concentrations of DPIN that might suggest dose-dependent changes in the efficiency of the absorption.

The second novelty of the present study is the description of the pharmacological effects of DPIN on both, glycemia and on the secretion of glucose homeostasis-related hormones in fasting healthy humans. Administration of 15 mg/kg of DPIN resulted in a pharmacological profile very similar to that found in preclinical rodent models [11]. D-Pinitol administration resulted in a small decrease in insulinemia at 90-, 120- and 180-min post DPIN intake, without affecting glycemia. The surprising stability of glycemia despite the reduction of insulin secretion was found in preclinical studies [11] to be mediated by the inhibition of liver pyruvate kinase, slowing glycolysis, and favoring the export of glucose to the blood circulation to sustain glycemia. Interestingly, insulin levels decreased 90, 120 and 180 min after oral intake of DPIN, and glucagon rises rapidly throughout the 3 h interval analyzed. Overall, the basal HOMA IR and HOMA β indexes resulting from these opposing pharmacological actions (Figure 4C,D) reflect stability in glucose homeostasis despite the net reduction in insulin secretion. This pharmacological profile of effects was explained in preclinical models by two mechanisms, a) a direct inhibitory action of DPIN on insulin secretion by pancreatic β-cells through modulation of ERK1/ERK2 signaling and b) the inhibitory action on insulin secretion exerted by acylated ghrelin (active form), that was enhanced by DPIN. In the absence of experimental data on the actions of DPIN in human islets or human hepatocytes, we cannot confirm this mechanism in humans- However, we did observe the increment of acylated ghrelin (the active form of the hormone) in humans subjected 60 min after the ingestion of DPIN. Interestingly, the effects of DPIN did not include a decrease of circulating free fatty acids (FFA) (Appendix A) consistent with direct insulin-mimetic action in the liver. Since glucagon promotes the release of free fatty acid in humans [19], blocking de novo synthesis of FFA, we interpret that DPIN sustains the release of FFA through the actions of glucagon, whose release is enhanced by the administration of this inositol. This finding may have a very interesting therapeutic potential in non-alcoholic steatohepatitis (NASH), where DPIN consumption might reduce liver steatosis and insulin resistance/diabetes type 2. Preliminary data recently published support this view [20]. Further studies are needed to demonstrate up to which point the actions of DPIN described in fasting humans are translated to better glucose homeostasis responses when a glucose tolerance test is performed combining oral administration of glucose and DPIN.

Regarding the effects of the 5 mg/kg dose, it is difficult to compare the changes in glucose homeostasis-associated plasma parameters (see Appendix A) between both doses of D-Pinitol. The critical time points in the higher DPIN dose were not evaluated in the lower DPIN dose. For the higher DPIN dose, glucose decreased during the first 30 min (not measured in lower DPIN dose) and insulin decreased at 90 and 180 (not measured in lower DPIN dose). Therefore, it is unknown how the lower dose would have responded at these certain time points. More research is needed to clarify which is the minimum dose required for the beneficial effects of DPIN in humans. 

The fact that DPIN administration resulted in enhanced Ghrelin release, together with the well-known effects of inositol on the gonadal axis [21,22], prompted us to examine the impact of this inositol on circulating levels of pituitary hormones. Surprisingly, DPIN, at the doses studied, only affected growth hormone (GH) secretion. The circulating concentrations of neither the gonadotrophins nor PRL or TSH were influenced by DPIN. The case of GH was intriguing since the rise of ghrelin, a GH secretagogue, and the drop of insulin that inhibits somatotroph release of GH (for review see [23,24,25]), might us expect a different profile of effects. However, experimental data in animal models suggest that DPIN might exert a potent insulin-mimetic action in the hypothalamus (acting on insulin receptor signaling cascade but not through a direct effect on insulin receptors) [12], and potentially in the somatotrophs, deregulating the hypothalamic control of GH via growth hormone release hormone (GHRH) and mimicking insulin on its direct inhibitory effect on pituitary GH secretion [24,25]. Whether this stimulatory effect of DPIN on hypothalamic insulin signaling happens in humans is still unknown, but it might explain the inhibitory action on GH levels. Interestingly, cortisol, a glucocorticoid that elevates glycemia, is reduced by DPIN, a fact that might reduce the need for additional pancreas insulin. However, in the absence of data evaluating the impact of DPIN on pituitary adrenocorticotrophin hormone (ACTH) secretion, we cannot identify whether the effects of DPIN on cortisol secretion are directly exerted in the adrenal gland or through disruption of hypothalamic/pituitary control of ACTH secretion. The lack of inhibitory effect of DPIN on FSH is also a relevant finding in the context of obesity and insulin resistance since this deficit in this hormone has been linked to NASH in populational studies [26].

Overall, the present study supports a protective role/preventive role derived from DPIN consumption on the endocrine pancreas, by reducing the load of insulin secretion and thus reducing one of the main factors contributing to insulin resistance. This is a novel complementary approach to the insulin-mimetic actions of D-Pinitol in the context of diabetes type 2 [27,28,29]. In addition, it suggests a preventive role in the development of liver steatosis, through the stimulatory role of glucagon secretion. However, several limitations demand more research to fully establish this hypothetical therapeutic role for dietary DPIN. It is important to understand the pharmacokinetics after repeated administration of this inositol, to establish adaptive changes on its metabolism/clearance. It is also very important to identify if the pharmacokinetics is influenced by other dietary elements, the existence of obesity/diabetes, and the presence of age-associated factors such as menopause. In any case, the present set of data establishes a framework for exploring the utility of this dietary ingredient.

### Limitations

Although the data obtained in the present study might help to establish the potential use of DPIN as a dietary strategy to fight against insulin resistance and diabetes type 2, (either alone or as an active ingredient in food preparations), there are several limitations to be considered. First, the small number of subjects might affect the evaluation of certain endocrine effects, especially affecting hormones with pulsatile secretion (i.e., gonadotrophins), or circadian rhythm-affected hormones (i.e., glucocorticoids). This is especially important for women, since we did not register the menstrual cycle phase of the female volunteers, a factor that must be considered in future experimental approaches. However, we considered that the sample size chosen is consistent with the goals of the study and enough to determine the pharmacokinetics parameters of oral administration of D-Pinitol, the main goal of the present investigation. Although the number of subjects and doses is limited and should be increased in future studies, it is remarkable that the pharmacological actions of D-Pinitol on the ghrelin-insulin-glucagon endocrine network-aligned with the results obtained from previous animal studies [11]. Another limitation is the need to increase the number of male and female subjects for having a clear power to differentiate potential sex-dimorphic responses. Finally, since D-Pinitol can be converted into D-chiroinositol by acid demethylation in the stomach, a study comparing the direct action of D-Pinitol with D-Chiroinositol has to be addressed.

## 5. Conclusions

The present study demonstrates that D-Pinitol, when given alone, is rapidly incorporated into the bloodstream, with a prolonged absorption period and a long half-life. Its co-administration with carbohydrates resulted in a partial reduction in the absorption, thus suggesting that the administration of this inositol might be optimal when given alone. D-Pinitol was found capable of reducing insulinemia while sustaining glycemia, through coordinated actions on glucagon and ghrelin secretion. This pharmacological profile suggests that the use of D-Pinitol as a dietary ingredient might be a reasonable strategy to help fight against the epidemics of insulin resistance/diabetes type 2 that has spread throughout most developed countries.

## Figures and Tables

**Figure 1 nutrients-14-04094-f001:**
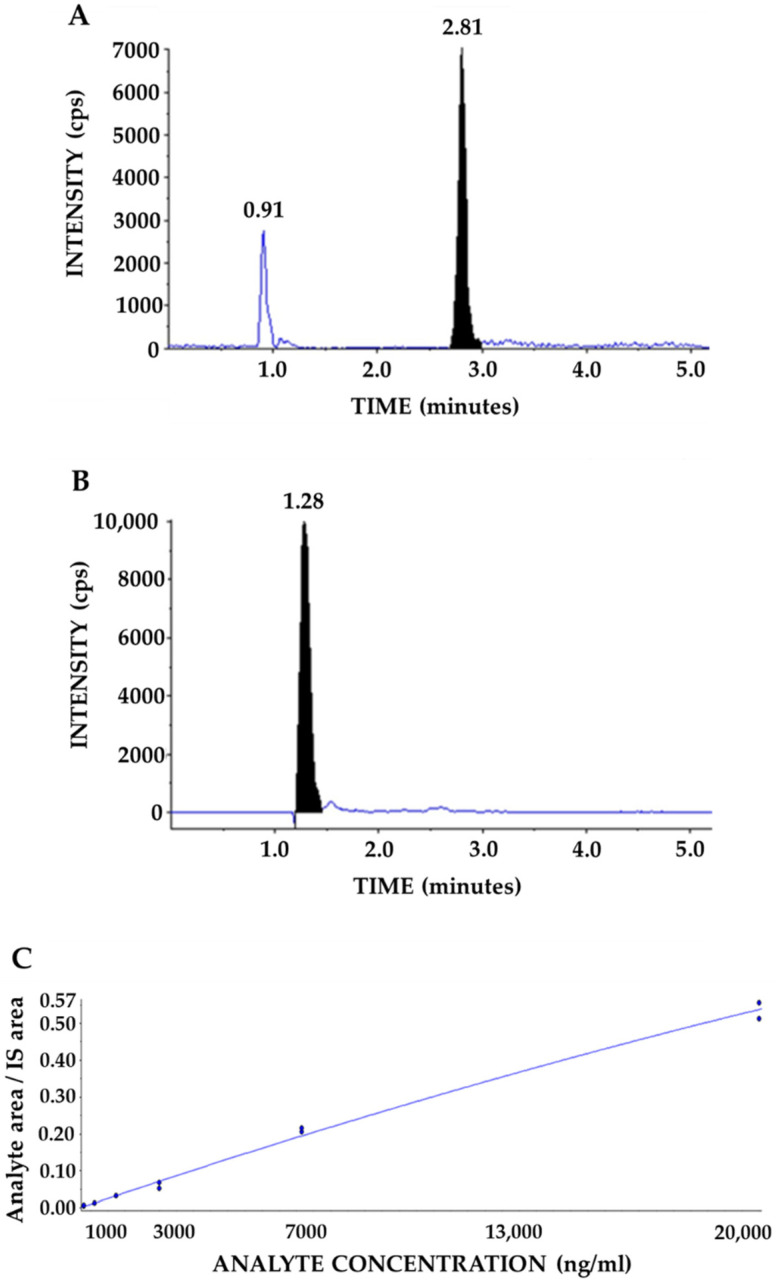
Chromatogram of D-Pinitol (**A**) and internal standard (**B**) in plasma sample in MRM. D-Pinitol calibration curve in plasma samples (Range 0.31–20 μg/mL) (**C**).

**Figure 2 nutrients-14-04094-f002:**
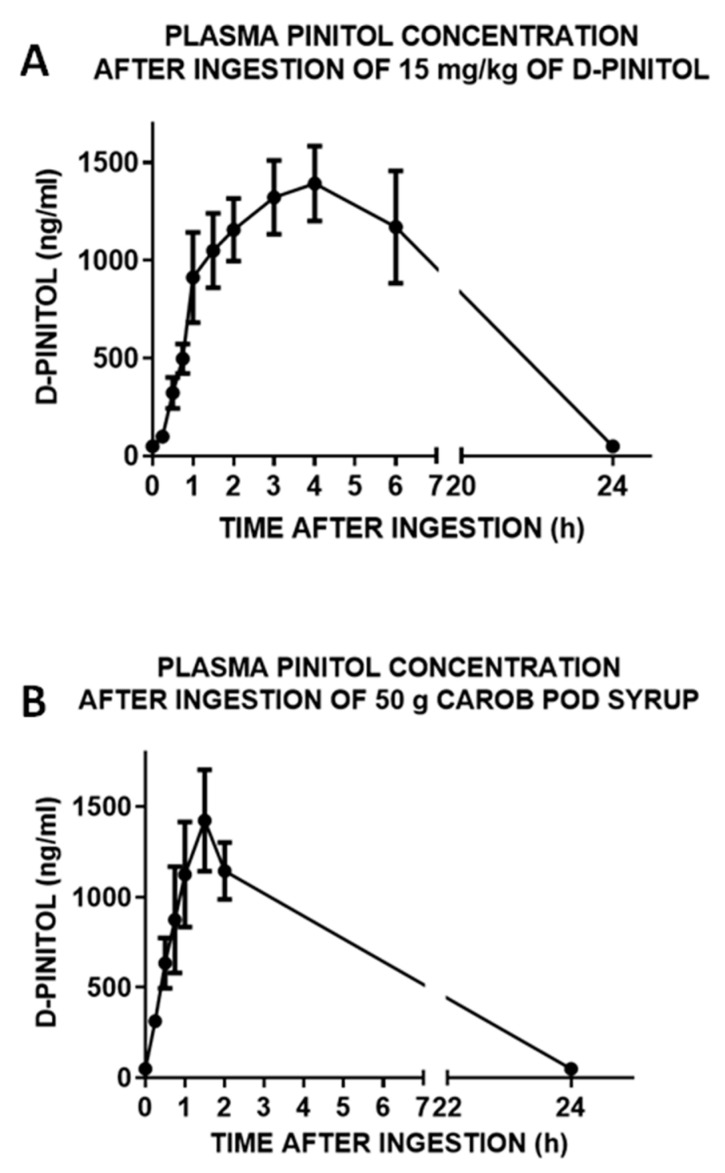
(**A**). Concentration of D-Pinitol in plasma (ng/mL) after an oral load (dose: 15 mg/kg body weight, N = 10 human subjects of both sexes, fasting for 12 h before the administration of D-Pinitol) at different times. The values are means ± SEM. (**B**). Concentration of D-Pinitol in plasma (ng/mL) after an oral load of 50 g of carbohydrates in a natural syrup (InnoSweet ^®^) containing 45.6% glucose, 47.3% fructose, 0.5% saccharose and 3.2% D-Pinitol (equivalent to a dose of D-Pinitol of 1600 mg or 22.8 mg/kg body weight in a person weighing 70 kg). Data are means ± SEM of 9 human subjects of both sexes, fasting for 12 h before the administration of the syrup.

**Figure 3 nutrients-14-04094-f003:**
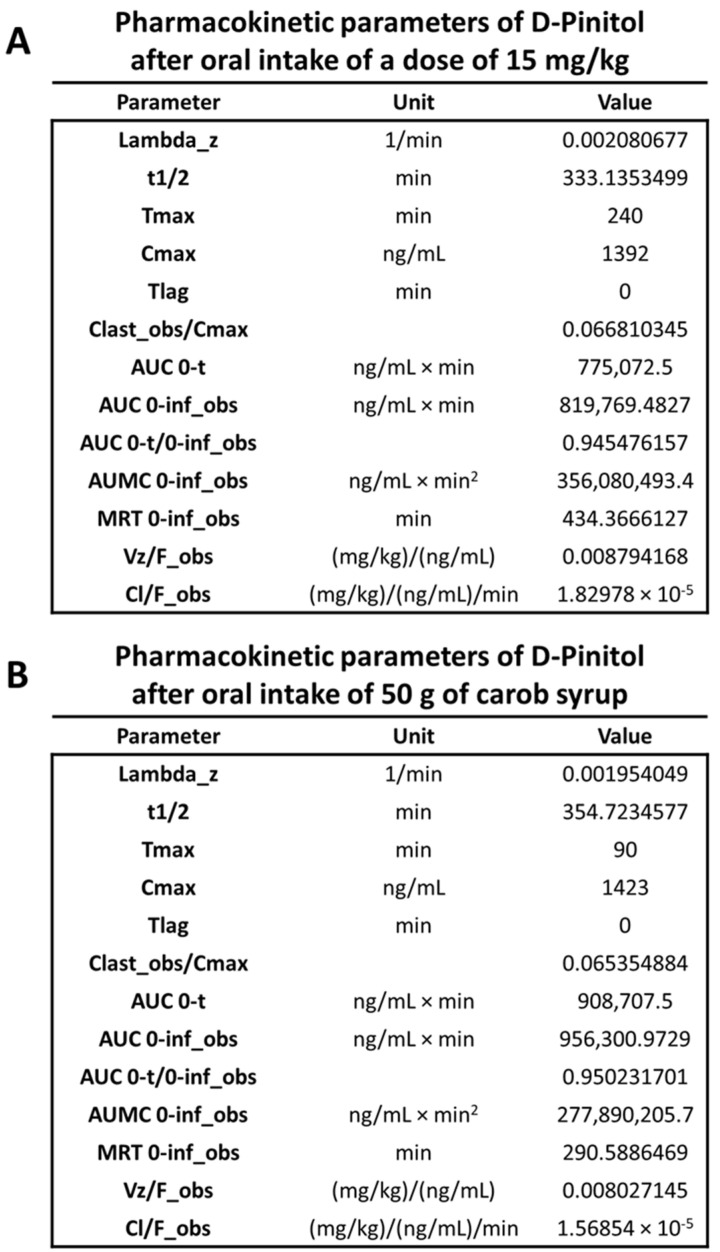
(**A**). Pharmacokinetic parameters calculated from the analysis of the concentration of D-Pinitol in plasma samples after a single oral dose of 15 mg/kg body weight, *n* = 10 human subjects of both sexes, fasting for 12 h before the administration of D-Pinitol. (**B**). Pharmacokinetic parameters calculated from the plasma concentration of D-Pinitol in plasma after an oral load of 50 g of carob pod syrup (InnoSweet ^®^) containing a dose of D-Pinitol of 1600 mg, *n* = 9 human subjects of both sexes, fasting for 12 h before the administration of the syrup. Lambda_z: first order rate constant associated with the terminal (log-linear) portion of the curve, estimated by linear regression of time vs. log concentration. t1/2: half-life. Tmax: time of maximum observed concentration. For non-steady-state data, the entire curve is considered. For steady-state data, Tmax corresponds to points collected during a dosing interval. If the maximum observed concentration is not unique, then the first maximum is used. Cmax: maximum observed concentration, occurring at Tmax. If not unique, then the first maximum is used. Tlag: extravascular input (model 200) only. Tlag is the time prior to the first measurable (non-zero) concentration. Cl: clearance. Clast_obs: total body clearance for extravascular administration. AUC: area under the curve. AUMC 0-inf_obs: area under the first moment curve (AUMC) extrapolated to infinity, based on the last observed concentration. MRT: mean residence time. Vz: Volume of distribution.

**Figure 4 nutrients-14-04094-f004:**
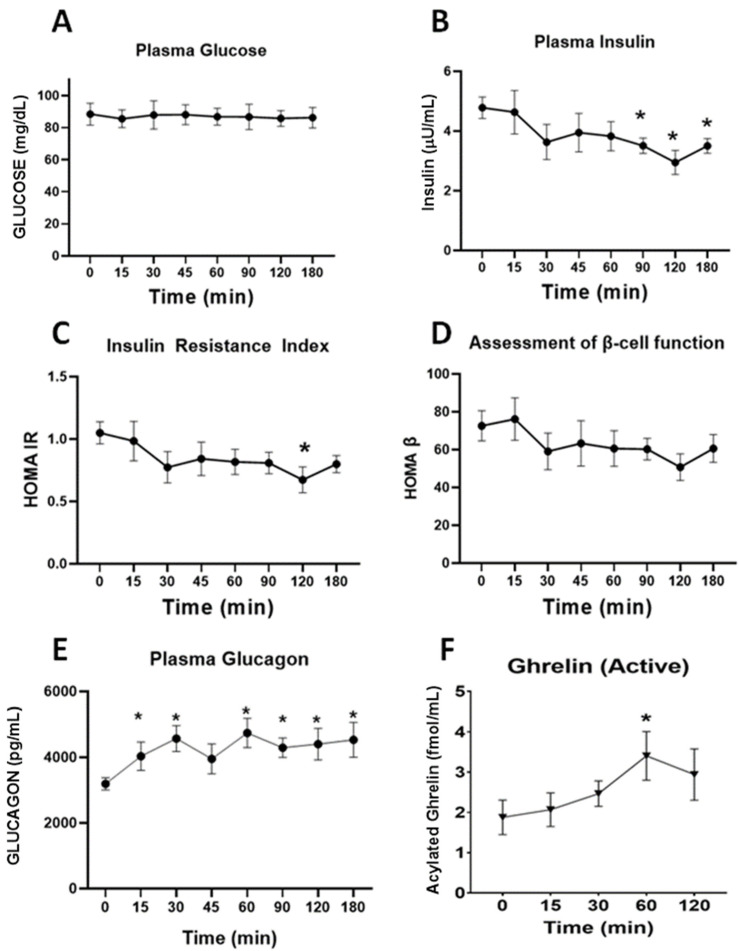
Changes in glucose homeostasis-associated plasma parameters in fasting human subjects receiving a single oral dose of D-Pinitol (15 mg/kg). The values are means ± SEM of 9–10 measures per group and time point. (**A**). Plasma glucose, (**B**). Plasma insulin, (**C**). Insulin Resistance Index (HOMA-IR), (**D**). Homeostatic assessment of β-cell function index (HOMA β), (**E**). Plasma glucagon (**F**). Plasma acylated ghrelin. Differences between groups were evaluated using one-way Anova + Fisher’s LSD test: * *p* < 0.05, vs. 0 time point.

**Figure 5 nutrients-14-04094-f005:**
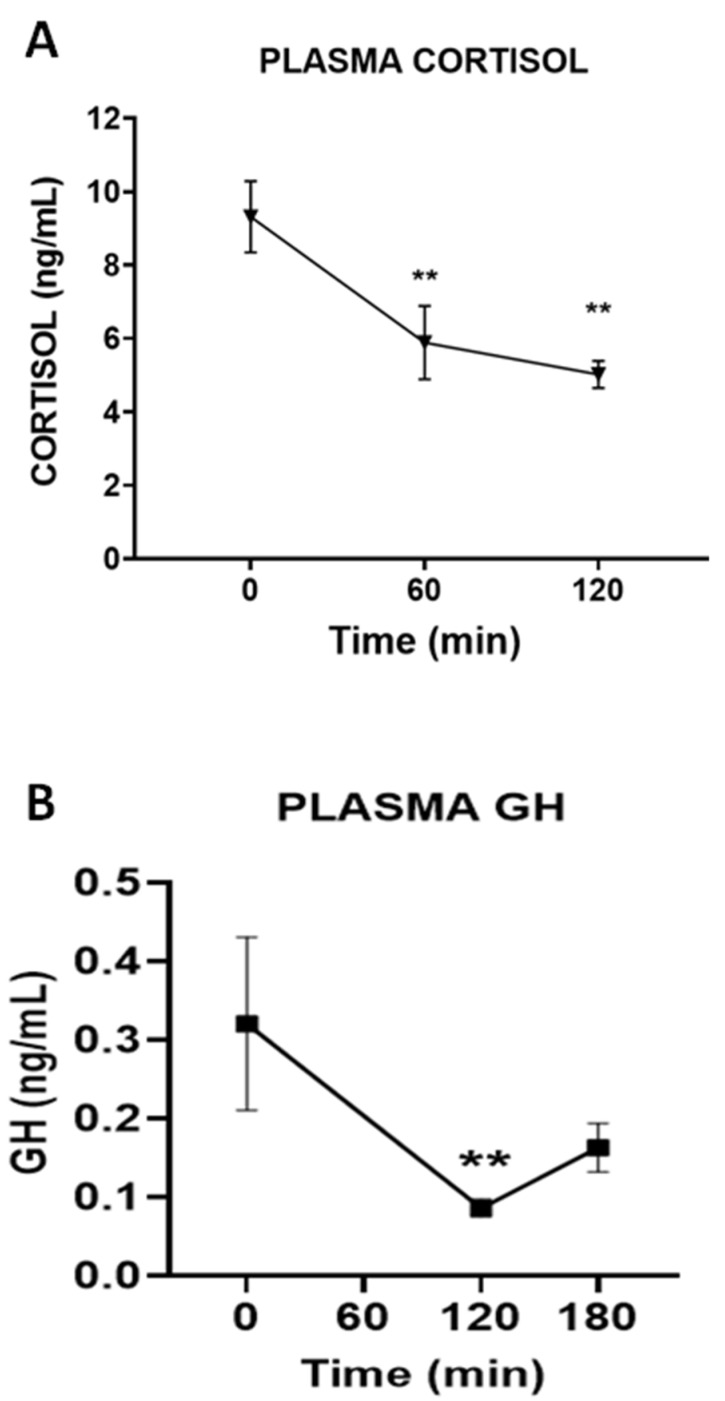
Changes in (**A**). Plasma growth hormone (GH), and (**B**). Plasma Cortisol after the ingestion of an oral dose of D-Pinitol (15 mg/kg), in human healthy volunteers. Differences between groups were evaluated using one-way Anova + Fisher’s LSD test: ** *p* < 0.01, vs. 0 time point.

**Figure 6 nutrients-14-04094-f006:**
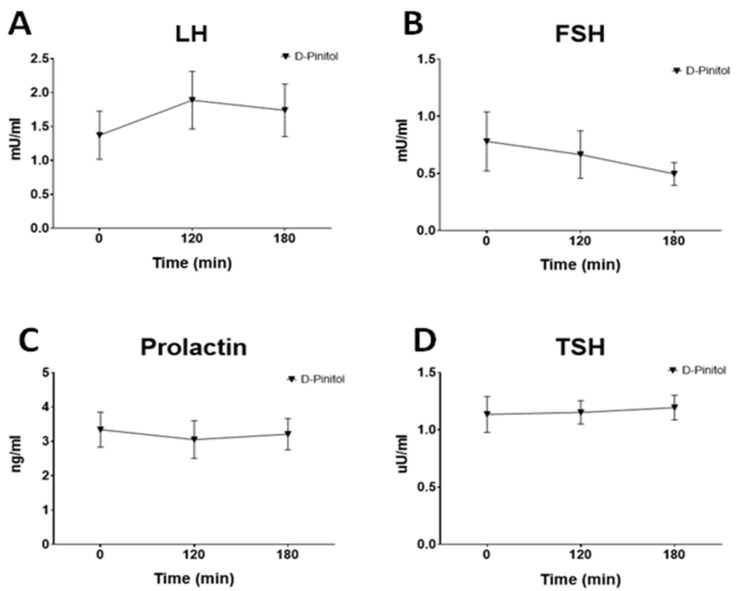
Changes in pituitary hormones plasma concentrations in fasting human subjects receiving a single oral dose of D-Pinitol (15 mg/kg). The values are means ± SEM of 9–10 measures per group and time point. (**A**). Plasma luteinizing hormone (LH), (**B**). Plasma Follicle-Stimulating Hormone (FSH), (**C**). Plasma Prolactin (PRL), (**D**). Plasma Thyrotropin (TSH). Differences between groups were evaluated using one-way Anova + Fisher’s LSD test.

**Table 1 nutrients-14-04094-t001:** Characteristics of human healthy volunteers.

Variables	Males	Females
*D-Pinitol sub-study* ^1^		
N	9	7
Age	41.5 ± 13.8	37.2 ± 14.8
Weight	84.6 ± 6.2	70.2 ± 16.2 (*)
Body Mass Index	28.1 ± 3.1	25.1 ± 4.9 (*)
Basal glucose (mg/dL)	82.5 ± 13.1	77.6 ± 9.6
Basal insulin (mIU/mL)	3.6 ± 1.2	3.5 ± 2.1
HOMA-IR	0.70 ± 0.44	0.71 ± 0.49
*Carob syrup sub-study*		
N	5	4
Age	37.7 ± 12.2	34.3 ± 11.6
Weight	84.6 ± 9.5	66.6 ± 12.4 (*)
Body Mass Index	27.2 ± 3.7	23.6 ± 4.0 (*)
Basal glucose (mg/dL)	81.8 ± 14.8	78.8 ± 8.4
Basal insulin (mIU/mL)	3.5 ± 1.4	2.95 ± 1.8
**HOMA-IR**	0.66 ± 0.28	0.58 ± 0.36

^1.^ Combined group of volunteers for doses of 15 and 5 mg/kg of D-Pinitol. (*) *p* < 0.05 males versus females, 2-way ANOVA.

## Data Availability

All data generated or analyzed during this study are available upon request by email to fernando.rodriguez@ibima.eu as a raw data file.

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
