# Peer review of "Pharmacokinetics and Endocrine Effects of an Oral Dose of D-Pinitol in Human Fasting Healthy Volunteers"

_nutrients, 2022, doi:10.3390/nu14194094_

Round 1

Reviewer 1 Report

1.     The title is inaccurate. The effects on endocrine control of glucose homeostasis were not evaluated in D-Pinitol with natural syrup, but pure D-Pinitol only. Because the glucose homeostasis and other hormones were not evaluated with D-Pinitol with natural syrup, it is unclear which source of pinitol (i.e., pure D-pinitol and/or pinitol with natural syrup) would be more beneficial as a dietary ingredient to help fight against insulin resistance/diabetes. Why was D-pinitol with natural syrup not evaluated for glucose homeostasis and other hormones?

2.     Why did the authors use a glucometer instead of a glucose analyzer instrument to measure glucose? The glucose measurements from the glucometer are not as accurate.

3.     The authors should consider combining Figures 2 and 3. It is easier to compare the pure D-Pinitol (15mg/kg) and D-Pinitol with natural syrup data if they are on the same graph and chart.

4.     This study has a very small sample size per group [Pure D-Pinitol 15mg/kg (n=10), Pure D-Pinitol 5mg/kg (n=6), and D-Pinitol with natural sugar (n=9)] and it is possible the study was underpowered. Please include in the statistical section the sample size estimation and power analysis for the research study.

5.     Lines 423-429. The study limitations should be clearly stated in the discussion. The limitations are stated like future research/direction.

6.     Figure S1. It is difficult to compare the changes in glucose homeostasis-associated plasma parameters between the pure D-Pinitol 5mg/kg and 15 mg/kg. The critical time points in the higher pinitol dose were not evaluated in the lower pinitol dose. For the higher pinitol dose, glucose decreased during the first 30 minutes (not measured in lower pinitol dose) and insulin decreased at 90 and 180 (not measured in lower pinitol dose). Therefore, it is unknown how the lower dose would have responded at these certain time points.

Minor

7.     References. This section should be numbered and coincide with the numbered citations in the manuscript.

8.     Abstract (lines 37 and 39). Cmax should be replaced with Tmax.

9.     Abstract. The results of cortisol were not included.

10.  Line 67 and 68. Write out INS-1 and clarify sentence (…insulin signaling “that” IPGs).

11.  Line 83. Clarify sentence (DPIN “is able of” enhancing glucose uptake).

12.  Line 92-94. List all study outcome variables.

13.  Line 107. Delete “is”

14.  Table 1. Should be labeled and age 36.6 years should include a “period and not a comma”.

15.  The methods section should include where pure D-Pinitol was extracted from. Was it purchased from a company?

16.  Gram should be abbreviated as “g” not “gr”

17.  Lines 161 and 262. There are different time points. Which one is correct?

18.  Figure 1C. It is difficult to read the numbers because they appear small and distorted.

19.  Line 277. Should it be “less variability” and not “more variability”.

20.  Line 325. Ghrelin doesn’t seem to be enhanced at 120 minutes after D-pinitol.

21.  Line 380. Should be 30 min post not 30’.

Reviewer 2 Report

Review on the manuscript “Comparative oral pharmacokinetics of D-Pinitol administered alone or as part of a natural syrup elaborated from carob pods in human healthy volunteers. Effects on endocrine control of glucose homeostasis” (nutrients-1833620).

The work has been done on an interesting topic and provides a plenty of data and findings worthy to add value into the field. However, there are critical concerns negatively affecting the validity of the work. The following comments are suggestable for improving the manuscript for the first round.

Major concerns:

1.      Although all statements throughout the manuscript are understandable, the manuscript is not written scientifically and harmonically. While a proofreading is necessary to be done, many statements/sentences should also be paraphrased using scientific grammars and expressions with an integrated writing style. More detailed comments in follow.

2.      In the first page: While the title does not represent the interesting aspects of the topic, it is highly confusing and too long as well. Abstract is long and not well-structured as well. I suggest to: summarize the first 3 sentences to one, add participants’ information, present the 3 study groups much clearer, present the results much more concise, and add a concluding general statement at the end. Keywords should also be written more wisely (e.g., not containing the terms presented in the title), if the authors would like to increase the chance of future citations.

3.      There are many phrases throughout the manuscript that can decrease the scientific level of inscription. (e.g., but not limited to, lines 87-88: “attractive” and “as a functional food”; lines 436: “thanks to”). In addition, in line 108, the term “studies” which has also been used frequently throughout the manuscript (or “study” elsewhere) is inappropriate and confusing and thus should be replaced with a better choice such as “sub-study”, “phase” or so. Moreover, the statement in line 108 is irrelevant to the section 2.1 and should be removed, as is also mentioned in the section 2.3.

4.      Table 1 should be presented based on the 3 study groups (not sex: males and females), and sex comparison can be presented in a differentiated rows or columns). It should also be clarified if there are any association between the 3 study groups regarding BMI and health-related data. The title of table should be revised accordingly (without mentioning “present study”).

5.      Section 2.2 (lines 152-154): Two different dosages of D-Pinitol (15 vs. 22.8 mg/kg) in comparison of the groups A and C. How can we persuade ourself that the study findings and the associated interpretations are valid and representative?

6.      Lines 144-148: it is mentioned that “subjects were randomly assigned to one of the following treatments …” but the distribution of participants is unequal (10, 6, and 9). A clear justification is needed to mention!

7.      The visual quality of Figure 1 is too low, and data are hard to recognize.

8.      The reason for compacting different irrelevant parts (A, B, and C) in Figure 2 (as well as A and B in Figure 3) is not clear. I would suggest to increase the number of figures or restructure them based on a reasonable justification. The long captions are also inappropriate.

9.      Considering the large amount of study variables and co-variables presented in the section Results, the section Discussion is too short, and a considerable part of study variables has not been discussed and interpreted. In addition, the Discussion needs further comparisons between the present findings and the comparable results from previous/available studies.

10.   At the final parts of Discussion, authors should also mention potential limitations of their study. In follow, a short paragraph concerning practical implications and applications should also be added, preferably accompanied by suggestions for future investigations without the limitations of the present study.

11.   The last major concern is about citation and referencing. Some statements throughout the manuscript should be supported by references (e.g., lines 55, 64, 76, and many more in Discussion). In addition, the reference list has not been numbered which made the 1st round of the review difficult.

Minor comments

12.   line 83: “L6” is confusing.

13.   Lines 96-99: The last statement of Intro seems to be more matched with the final parts of Discussion or Conclusions.

14.   Lines 175 and 194 and 200: subheadings should be used.

15.   Lines 242-244: the section 2.7 should be removed, as is presented in the relevant part, lines 464-465.

16.   Lines 277-284: the statements seem to be related to the section Discussion.

Regards,

23 July 2022

Author Response

Please, see attachment

Round 2

Reviewer 1 Report

1.     Please include in the statistical section the sample size estimation and power analysis for the research study (lines 132-146 should be summarized and moved to the statistical section).

2.     Table 1. Results from the table should be briefly mentioned in the results section.

3.     Study Design. The aims should not be included in the study design, only the last paragraph of the introduction. This section should only address the study design.

4.     It is difficult to compare the changes in glucose homeostasis-associated plasma parameters between the pure D-Pinitol 5mg/kg and 15 mg/kg. The critical time points in the higher pinitol dose were not evaluated in the lower pinitol dose. For the higher pinitol dose, glucose decreased during the first 30 minutes (not measured in lower pinitol dose) and insulin decreased at 90 and 180 (not measured in lower pinitol dose). Therefore, it is unknown how the lower dose would have responded at these certain time points. NOT ADDRESSED IN THE DISCUSSION—Authors should consider adding to the discussion a statement regarding it being unknown if 5mg/kg had an effect on glucose homeostasis because critical timepoints were not included (can also be included as a study limitation—can’t compare the effect of 15mg/kg vs 5mg/kg on glucose homeostasis). 

5.     Line 393. The pharmacological profile of DPIN-5mg/kg was not reported; therefore, this statement is inaccurate.

6.     Lines 391-413. Need to include that the study did not evaluate DPIN with sugar on glucose homeostasis.

7.     Need to mention the insulin resistance index in the discussion.

8.     Abstract (lines 37 and 39). Cmax should be replaced with Tmax. DID NOT ADDRESS (now lines 31-32)

9.     Line 92-94. List all study outcome variables. PARTIALLY ADDRESSED (need to add FFA)

10.  The methods section should include where pure D-Pinitol was extracted from. Was it purchased from a company? DID NOT ADDRESS (need to put this information in the study design section of the manuscript)

11.  Gram should be abbreviated as “g” not “gr”. NOT ADDRESSED (need to review the entire manuscript)

12.  Lines 161 and 262. There are different time points. Which one is correct? DID NOT ADDRESS (now lines 170 and 271)

13.  Line 325. Ghrelin doesn’t seem to be enhanced at 120 minutes after D-pinitol. DID NOT ADDRESS (now lines 325)

14.  Line 380. Should be 30 min post not 30’. DID NOT ADDRESS (now line 395)

15.  Line 102. Should be ingestion, not injection

16.  Line 103. Should be to, not tp

17.  Table 1. All variables need units, if appropriate. DPIN subgroup should indicate that it is a combined subgroup of 15mg/kg and 5mg/kg.

18.  Line 174. Include ghrelin and insulin resistance index; What about cortisol and GH?

19.  Line 157; 368-369. Sentence needs to be revised for clarity

20.  Line 172. Need to make a clear distinction between the different doses of DPIN.

21.  Line 298. Update figure

22.  Figure 4A. Should an asterisk be added to glucose at minute 30 since the decrease at this timepoint was significant?

23.  Why is the Plasma GH figure not included with the other pituitary hormone figures?

24.  Line 452. Word misspelled (should be even though, not even tough)

Author Response

ANSWER TO REFEREE 1

We have addressed all changes suggested by te referee,

  1. Please include in the statistical section the sample size estimation and power analysis for the research study (lines 132-146 should be summarized and moved to the statistical section).

ANSWER:  We have moved and corrected the simple size calculation to the statistical section as requested. A brief summary was placed in the study design as suggested.

  1. Table 1. Results from the table should be briefly mentioned in the results section.

ANSWER: We thank the referee for the suggestion. A new section 3.1 describing the main characteristics of the recruited volunteers has been incorporated to the manuscript.

  1. Study Design. The aims should not be included in the study design, only the last paragraph of the introduction. This section should only address the study design.

ANSWER: We have moved the aims paragraph to the introduction, as suggested.

  1. It is difficult to compare the changes in glucose homeostasis-associated plasma parameters between the pure D-Pinitol 5mg/kg and 15 mg/kg. The critical time points in the higher pinitol dose were not evaluated in the lower pinitol dose. For the higher pinitol dose, glucose decreased during the first 30 minutes (not measured in lower pinitol dose) and insulin decreased at 90 and 180 (not measured in lower pinitol dose). Therefore, it is unknown how the lower dose would have responded at these certain time points. NOT ADDRESSED IN THE DISCUSSION—Authors should consider adding to the discussion a statement regarding it being unknown if 5mg/kg had an effect on glucose homeostasis because critical timepoints were not included (can also be included as a study limitation—can’t compare the effect of 15mg/kg vs 5mg/kg on glucose homeostasis). 

ANSWER: We appreciate the comments. The study of the 5 mg/kg dose was performed to compare the rate of absorption with that of the high D-Pinitol dose. The referee is right, and we have translated his/her comment to the discussion section.

  1. Line 393. The pharmacological profile of DPIN-5mg/kg was not reported; therefore, this statement is inaccurate.

ANSWER: we removed the reference to the dose of 5 mg/kg

  1. Lines 391-413. Need to include that the study did not evaluate DPIN with sugar on glucose homeostasis.

ANSWER: we have modified the text accordingly and included a text explicitly indicating that : “Further studies are needed to demonstrate up to which point the actions of DPIN described in fasting humans are translated to better glucose homeostasis responses when a glucose tolerance test is performed combining oral administration of glucose  and DPIN.”

  1. Need to mention the insulin resistance index in the discussion.

ANSWER: a mention to the data of Figure 4C is now placed in the discussion after the insulin data description (paragraphs of lines 391-414)

  1. Abstract (lines 37 and 39). Cmax should be replaced with Tmax. DID NOT ADDRESS (now lines 31-32)

ANSWER: We refer now to both, to the time at which the maximal concentration is reached (Tmax) and to the differences in the absolute value of the maximal concentration (Cmax). Both parameters are important to understand the PK: . “Data indicate that when given alone, the oral absorption of D-Pinitol is of extended duration, with a Tmax reached after almost 4 hours, and a half-life greater than 5 hours. When the source of D-Pinitol was a carob pods-derived syrup, Cmax was reduced to a 40% of the expected based on the data of D.Pinitol alone, suggesting a reduced absorption probably because of competition with monosaccharide transport. In this group, Tmax was reached before than that of D-pinitol alone, but the estimated half-life remained the same.”

  1. Line 92-94. List all study outcome variables. PARTIALLY ADDRESSED (need to add FFA)

ANSWER: we completed the list as requested.

  1. The methods section should include where pure D-Pinitol was extracted from. Was it purchased from a company? DID NOT ADDRESS (need to put this information in the study design section of the manuscript)

ANSWER: D-Pinitol was extracted and purified by EURONUTRA SL. All the information on this product is now in the methods section. “D-Pinitol (3 O methyl d chiro-inositol, 98% purity) was generously provided by Euronutra (https://www.euronutra.com/, Málaga, Spain)”

  1. Gram should be abbreviated as “g” not “gr”. NOT ADDRESSED (need to review the entire manuscript)

ANSWER: Done

  1. Lines 161 and 262. There are different time points. Which one is correct? DID NOT ADDRESS (now lines 170 and 271)

ANSWER: We gratefully acknowledge the identification of this error. The correct sampling times are 0, 15, 30, 45, 60, 90, 120, 180, 240, 360 and 1440 min after D-Pinitol oral administration.

  1. Line 325. Ghrelin doesn’t seem to be enhanced at 120 minutes after D-pinitol. DID NOT ADDRESS (now lines 325)

ANSWER: we corrected the description. Now it reads: Acylated Ghrelin was found to be enhanced 60 after DPIN ingestion.

  1. Line 380. Should be 30 min post not 30’. DID NOT ADDRESS (now line 395)

ANSWER: corrected

  1. Line 102. Should be ingestion, not injection

ANSWER: corrected

  1. Line 103. Should be to, not tp

ANSWER: corrected

  1. Table 1. All variables need units, if appropriate. DPIN subgroup should indicate that it is a combined subgroup of 15mg/kg and 5mg/kg.

ANSWER: corrected

  1. Line 174. Include ghrelin and insulin resistance index; What about cortisol and GH?

ANSWER: we have described the timing of sampling for all hormones

  1. Line 157; 368-369. Sentence needs to be revised for clarity

ANSWER: We have modified that sentence. Now it reads: “The present study includes sampling times (6 and 24 h post-DPIN ingestion) that were not evaluated in previous studies monitoring plasma DPIN concentrations after oral intake. Those studies limited the analysis of DPIN concentrations to the first 4 h post-ingestion [10, 13].

  1. Line 172. Need to make a clear distinction between the different doses of DPIN.

ANSWER: corrected

  1. Line 298. Update figure

ANSWER: corrected

  1. Figure 4A. Should an asterisk be added to glucose at minute 30 since the decrease at this timepoint was significant?

ANSWER: corrected

  1. Why is the Plasma GH figure not included with the other pituitary hormone figures?

ANSWER:  GH was not measured in the lower dose.

  1. Line 452. Word misspelled (should be even though, not even tough)

ANSWER: corrected

Reviewer 2 Report

Although the authors’ efforts are appreciated and the manuscript has had some improvements, there are still a number of flaws and concerns which cannot be scientifically ignored. From the comments that I suggested for improvement of the manuscript for the first round, the authors addressed only some of them with an acceptable rate, and the majority of them are hardly acceptable or unacceptable scientifically. In addition, the current yellow-highlighted parts don’t display the detailed actions taken by authors; as a standard procedure in scientific revision, it is expected that authors should use “Track Changes” for highlighting the revisions, and to point out each revision in the cover letter by line numbers and full explanations. Furthermore, the inscription of paper still has some considerable flaws in both structure and language. Unfortunately, I am unable to persuade myself to accept the current version of the manuscript. From my viewpoint, the current version of the manuscript is far from the Nutrients standards. However, if the editors would like to keep this paper in the cycle, I would suggest authors to spend more time to conduct a more detailed revision and provide point-by-point actions/clarifications for the previous comments in both manuscript and cover letter.

Regards,

13 August 2022; 9:56 AM

Author Response

Comments and Suggestions for Authors

Second Revision

Although the authors’ efforts are appreciated and the manuscript has had some improvements, there are still a number of flaws and concerns which cannot be scientifically ignored. From the comments that I suggested for improvement of the manuscript for the first round, the authors addressed only some of them with an acceptable rate, and the majority of them are hardly acceptable or unacceptable scientifically.

ANSWER: We appreciate referee’s comments, but such a criticism should be as detailed as the referee request for our response. We described the changes on the manuscript on referee’s 1 report with the level of precision that the referee requested. And we also described how we addressed the queries proposed by the present reviewer. If a revision is not scientifically acceptable, as the referee suggests, It would be necessary to know on which basis this criticism is formulated. In order to facilitate the re-evaluation of our response, we have translated to the present document the main changes requested by the referee.

In addition, the current yellow-highlighted parts don’t display the detailed actions taken by authors; as a standard procedure in scientific revision, it is expected that authors should use “Track Changes” for highlighting the revisions, and to point out each revision in the cover letter by line numbers and full explanations.

ANSWER: We understand the request of the referee. But when the requested changes affect wide sections of the manuscript, the use of “Track Changes” generates a mess of text marks and modifications. We did the highlighting procedure because it respects the final appearance of the manuscript and helps the reader to localize the changes made. We feel that only minor changes should be presented on “tracking mode” to simplify the reading. In the current version we have modified only the minor changes suggested by referee 1, and the reader can see how it complicates the examination of the text. Anyhow, in the present response, and when possible, we have detailed the changes made in the first revision, giving the line numbers to facilitate referee’s examination.

Furthermore, the inscription of paper still has some considerable flaws in both structure and language. Unfortunately, I am unable to persuade myself to accept the current version of the manuscript. From my viewpoint, the current version of the manuscript is far from the Nutrients standards. However, if the editors would like to keep this paper in the cycle, I would suggest authors to spend more time to conduct a more detailed revision and provide point-by-point actions/clarifications for the previous comments in both manuscript and cover letter.

ANSWER: We provide below the detailed  description of the changes suggested by the referee.

First  Revisión

Review on the manuscript “Comparative oral pharmacokinetics of D-Pinitol administered alone or as part of a natural syrup elaborated from carob pods in human healthy volunteers. Effects on endocrine control of glucose homeostasis” (nutrients-1833620).

The work has been done on an interesting topic and provides a plenty of data and findings worthy to add value into the field. However, there are critical concerns negatively affecting the validity of the work. The following comments are suggestable for improving the manuscript for the first round.

 Major concerns:

  1. Although all statements throughout the manuscript are understandable, the manuscript is not written scientifically and harmonically. While a proofreading is necessary to be done, many statements/sentences should also be paraphrased using scientific grammars and expressions with an integrated writing style. More detailed comments in follow.

We have carefully edited the manuscript following referee’s suggestions, using the scientific rigor that the experimental research in human demands. With respect to the grammar/scientific style of writing, we do not understand referee’s complains, since we attach to an objective experimental description, avoiding subjective biass that might modify the potential scientific relevance of the findings reported. It is true that we are not native English speakers, but the critic posed herewith should be accompanied by a detailed description of the writing weaknesses that must be improved.

  1. In the first page: While the title does not represent the interesting aspects of the topic, it is highly confusing and too long as well. Abstract is long and not well-structured as well. I suggest to: summarize the first 3 sentences to one, add participants’ information, present the 3 study groups much clearer, present the results much more concise, and add a concluding general statement at the end. Keywords should also be written more wisely (e.g., not containing the terms presented in the title), if the authors would like to increase the chance of future citations. 

ANSWER: We have modified the title than now reads:

Pharmacokinetics and endocrine effects of an oral dose of D-Pinitol in human healthy volunteers

This new title focuses the interest of the work on the pharmacokinetics of D-Pinitol in healthy human subjects, which is the key research aim of the report. Regarding the Abstract and key words, they were modified following Referee’s recommendation, but keeping all the essential information and the Nutrients style recommendations. A concluding remark states: “As a conclusion, D-Pinitol is absorbed through the oral route, having an extended half-life and displaying the pharmacological profile of an endocrine pancreas protector, of potential interest for the Treatment or prevention of insulin resistance-associated conditions”. (Lines 43-47)

  1. There are many phrases throughout the manuscript that can decrease the scientific level of inscription. (e.g., but not limited to, lines 87-88: “attractive” and “as a functional food”; lines 436: “thanks to”). In addition, in line 108, the term “studies” which has also been used frequently throughout the manuscript (or “study” elsewhere) is inappropriate and confusing and thus should be replaced with a better choice such as “sub-study”, “phase” or so. Moreover, the statement in line 108 is irrelevant to the section 2.1 and should be removed, as is also mentioned in the section 2.3.

We have followed referee’s suggestions and modified the manuscript, specially methodology and results accordingly. The expressions highlighted by the referee have been removed from the manuscript. Statement in line 108 was deleted as suggested.

  1. Table 1 should be presented based on the 3 study groups (not sex: males and females), and sex comparison can be presented in a differentiated rows or columns). It should also be clarified if there are any association between the 3 study groups regarding BMI and health-related data. The title of table should be revised accordingly (without mentioning “present study”).

We have modified the title of the table (line 193) and the presentation of the participants on the basis of the compound received, either D-Pinitol (combining the subjects of the doses of 15 and 5 mg/kg) or Syrup). We have done regression analysis on the basis of weight or BMI versus Cmax. There were no correlations of D‐Pinitol Cmax with BMI (r2=0.44, p (two tailed) = 0.05). Mean peak plasma D‐Pinitol values were similar in females (1527 + 589 ng/ml) than in male subjects (1562 + 178 ng/ml). This data can be found in new section 3.3., lines 356-358.

  1. Section 2.2 (lines 152-154): Two different dosages of D-Pinitol (15 vs. 22.8 mg/kg) in comparison of the groups A and C. How can we persuade ourself that the study findings and the associated interpretations are valid and representative?

We thank the referee for the comment. The use of 5mg/kg in a sub-study was designed basically to see how efficient is the absorption with the low dose, but as both referee’s state, we lacked critical time points. So we have moved the entire set of data to a new Supplementary Figure 1, and included a discussion supporting referee’s view. We have modified the methodology to fully describe this fact .

In the  study design section (lines 157-161) Concerning the sample size, we calculated it using the Gpower program, version 3.1.9.2., taking in consideration that the expected variations in plasma D‐Pinitol concentrations might exceed 3‐fold in between time points. Recruitment of male and female

160 volunteers was done to reach the calculated sample size, as described in Table 1 and statistics section.

In the statistics section (Lines 281-290): “The main variable of the study were the Cmax and the area under the curve (AUC) obtained when monitoring variations of plasma D-Pinitol levels at different times after oral intake. Since plasma concentrations at the different time points selected after oral intake of D-Pinitol can display more than a 3-fold variation, we selected a restrictive size of effect of 2. That gives a sample size of 7 subjects per group for comparing D-Pinitol levels versus time after ingestion of either D_Pinitol or Carob Syrup. Because we wanted to have a representation of men and woman, we increased the numbers to 10 and 9 respectively. For the dose of 5 mg/kg we could only recruit 6 out of the 7 subjects, so we decided to present these data as supplementary information, but we did not perform a full analysis with this sub-study. Thus, we recruited a total of 25 subjects.”

  1. Lines 144-148: it is mentioned that “subjects were randomly assigned to one of the following treatments …” but the distribution of participants is unequal (10, 6, and 9). A clear justification is needed to mention!

We have detailed the procedures for sample size and the distribution in the previous question. The size was estimated on the basis of g-power calculation and the distribution based on the aleatory selection of candidates for each group.

  1. The visual quality of Figure 1 is too low, and data are hard to recognize. 

Figure 1 has been modified and resized to improve quality

  1. The reason for compacting different irrelevant parts (A, B, and C) in Figure 2 (as well as A and B in Figure 3) is not clear. I would suggest to increase the number of figures or restructure them based on a reasonable justification. The long captions are also inappropriate.

Figures 2 & 3 have been modified following the sugestion of both referees. New Figure 2 now shows the comparative pharmacokinetic curves of plasma D-Pinitol from volunteers receiving the pure compound at a dose of 15mg/kg and from volunteers that were given D-pinitol as part of a carob pod syrup. New Figure 3 displays the comparative PK analysis data. New Figure 4 displays data on glucose, free fatty acids, pancreatic hormones and ghrelin, and New Figure 5 the plasma concentrations of cortisol and growth hormones. Finally, new Figure 6 displays the plasma concentrations of pituitary hormones.

  1. Considering the large amount of study variables and co-variables presented in the section Results, the section Discussion is too short, and a considerable part of study variables has not been discussed and interpreted. In addition, the Discussion needs further comparisons between the present findings and the comparable results from previous/available studies.

      We have extended the discussion as suggested, covering all the different comparison that were not addressed, as suggested by the referee, discussing aspects on PK  and doses (lines 424-455), effects on hormones modulating glycaemia (Lines 456-475, 488-495), free fatty acids (476-484),  and  pituitary hormones and cortisol 496-516). A limitation section has been included (lines 530-545)  highlighting the need of additional research for understanding the physiological impact of D-Pinitol administration in humans.

  1. At the final parts of Discussion, authors should also mention potential limitations of their study. In follow, a short paragraph concerning practical implications and applications should also be added, preferably accompanied by suggestions for future investigations without the limitations of the present study.

                  A new section of limitations of the study (Lines 530-545) has been incorporated to the end of the discussion, which include also recommendations for future research.

“The present study has several limitations. First, the small number of subjects might affect the evaluation of certain endocrine effects, specially affecting hormones with pulsatile secretion (i.e. gonadotrophins), or circadian rhythm-affected hormones (i.e. glucocorticoids). Higher number and different timing of administration should be considered. Even tough, we considered that the sample size chosen is consistent with the goals of the study and enough to determine the effect of expected magnitude of scientific significance. In fact, even with this limitation, it is notable that the findings described in this study strongly supports the results obtained from our previous animal studies [11].  Another limitation is the need to increase the numbers for having a clear power to differentiate potential sex-dimorphic responses. Finally, since D-Pinitol can be converted into D-chiroinositol by acid demthylation in the stomach, a study comparing the direct actions of both, D-pinitol and D-chiroinositol, has to be addressed.”

  1. The last major concern is about citation and referencing. Some statements throughout the manuscript should be supported by references (e.g., lines 55, 64, 76, and many more in Discussion). In addition, the reference list has not been numbered which made the 1stround of the review difficult.

      We numbered and modified the references accordingly, keeping in mind that the scarce number of studies on D-Pinitol in humans, and the limited focus of a pharmacokinetic study in humans.

Minor comments 

  1. line 83: “L6” is confusing. L6 refers to a particular myoblast cell line, as described now in the text (line 85-86)
  2. Lines 96-99: The last statement of Intro seems to be more matched with the final parts of Discussion or Conclusions.

We have moved this last statement to the discussion as requested.

  1. Lines 175 and 194 and 200: subheadings should be used.

2.4.1 MS method.

2.4.2 Sample preparation.

2.4.3 Pharmacokinetic study.

We have modified the the subheading accordingly to the  Nutrients Style.

  1. Lines 242-244: the section 2.7 should be removed, as is presented in the relevant part, lines 464-465.

We have removed this section, following referee’s recommendation

  1. Lines 277-284: the statements seem to be related to the section Discussion.

 This section has been moved to Discussion
